# Design and Performance Analysis of Lamina Emergent Torsional Joints Based on Double-Laminated Material Structure

Buchuan Ma [1], Lifang Qiu [1], Beiying Liu [1], Yue Yu [1], Ningning Liu [1] and Guimin Chen [2,*]

[1] School of Mechanical Engineering, University of Science and Technology Beijing, 30 Xueyuan Road, Haidian District, Beijing 100083, China; mabuchuan123@126.com (B.M.); qlf@ustb.edu.cn (L.Q.); byliu@me.ustb.edu.cn (B.L.); yuyueustb@163.com (Y.Y.); liuningning_0412@163.com (N.L.)

[2] State Key Laboratory for Manufacturing Systems Engineering and Shaanxi Key Laboratory of Intelligent Robots, Xi'an Jiaotong University, Xi'an 710049, China

[*] Correspondence: guimin.chen@xjtu.edu.cn

**Featured Application: A novel double-laminated Lamina Emergent Torsional joint (DL-LET) is described here and shows promise in applications where both high flexibility and high accuracy are required. It is beneficial for joints in microrobots due to its initial planar state and large deflection range. The flexibility of DL-LET can also offer safe interaction and self-adaption for robots.**

**Abstract:** Flexibility and accuracy are two key aspects of the performances of compliant joints. In order to obtain high flexibility while maintain high accuracy, this paper proposes a design method to improve the tensile stiffness of Lamina Emergent Torsional (LET) joint by utilizing double-laminated material structure. The joint is made of a LET joint and a layer of flexible H18 aluminum foil fixing on it (called double-laminated LET, DL-LET). The kinetostatic model for the joint is given, and the equations needed to calculate the equivalent spring constant are derived. The model is verified using finite element analysis (FEA). The results obtained through two different ways coincide with each other very well. The results show that DL-LET and LET joints have similar bending stiffness, while the tensile stiffness of the DL-LET joint is much larger than that of the LET joint. The results are validated by tensile tests. Finally, to further demonstrate the extension of this idea, a DL-Triple-LET joint is presented and compared to the Triple-LET joint.

**Keywords:** compliant mechanism; Lamina Emergent Torsional (LET) joint; double-laminated material structure; tensile stiffness; double-laminated Triple-LET

## 1. Introduction

Traditional mechanisms use articulated joints to connect their rigid body elements and to achieve their motion, while compliant mechanisms obtain their motion through the deformation of the compliant joints [1,2]. Compliant mechanisms have many advantages over traditional mechanisms, such as no backlash or friction, higher precision, and reliability, reduced number of components, and assembly time [3,4]. Lamina emergent mechanisms (LEMs) are a kind of compliant mechanisms that are manufactured from sheet goods with motion out of the fabrication plane [5]. Researchers have designed many joints for LEMs. Finding suitable compliant joints [6] can be a key step in designing LEMs. Jacobsen et al. [7] first proposed the design concept of utilizing torsional deflection to obtain out-of-plane motions from planar structures, leading to a new group of compliant joints for LEMs called lamina emergent torsional (LET) joints. The LET joints and their equivalent stiffness models were presented in Ref. [8]. Although LET joints have been used in many LEMs [9,10] due to their planar configurations and large displacement ranges, they yield considerable parasitic motions when subject to tensile or compression loads. In order to solve this problem, three joints were introduced in [11]: Inverted Lamina Emergent joint (I-LEJ), Tension Lamina Emergent joint (T-LEJ), and Inverted tension Lamina Emergent joint (IT-LEJ). They

demonstrate higher stiffness in tensile or compression loads conditions, at the cost of limited bending flexibility compared to the LET joint and requiring more material area. Merriam et al. presented compound joints [12] and two lattice flexure types, called X-type and V-type [13], which can reduce bending stiffness and be validated through physical prototypes. Ref. [14] proposed the shape factor of compliant arrays (CA) combined with the material and geometrical properties. Moreover, the shape factor is applied to the analysis of two special CA (Unidirectional one-way flexible array and two-way flexible array), which effectively showed the balance between elastic bending and strength. Qiu et al. [15] designed a Triple-LET flexure joint to achieve larger deformation, and an application on the slider-crank mechanism was finished. Ref. [16] presented an Outside-Deployed Lamina Emergent Joint OD-LEJ (OD-LEJ) with a smaller equivalent spring constant. OD-LEJ can achieve nearly 170.17° of rotation without plastic deformation, and it can do better than LET joint in large angular displacement. In terms of characteristic parameters, Ref. [17] designed a double C-type flexure hinge and proposed a method to remove some materials of the flexure hinges in certain rules, which ensured anti-tensile property and improved bending property. The performance of the double C-type flexure hinges was compared through the finite element simulation analysis. Chen et al. [18] introduced a type of LET joint called membrane-enhanced LET joints (M-LET joints). M-LET can maintain large angular displacements and mitigate parasitic motions and low buckling resistance. Several different mechanisms are proposed to describe the M-LET joint better. Recently, some LET joints with Spatial motion have also been developed that provide more possibilities for applying LEMs [19–21]. With the development of technology, some new analysis methods of compliant mechanisms with Spatial motion have gradually emerged, such as the spatial Beam constraint model (BCM) [22], the Spatial Chained beam-constraint-model (CSBCM) [23], or the Chained Power Series Model [24], etc.

In terms of application, Olsen et al. [25] used rigid-body replacement to synthesize a LEM Audi A4 cup holder. Wilding et al. [26] introduced spherical LEMs and discussed the fundamentals of spherical LEMs. The paper showed different spherical 4R LEMs, and classification was made on spherical 4R LEMs. Dodgen et al. [27] described a spinal implant based on the LET joint, which can provide a customizable nonlinear stiffness in multiple directions. Noveanu et al. [28] proposed an analytical compliance-based matrix method to design and analyze a novel displacement amplified gripper with right circularly corner-filleted flexure hinges. Lobontiu [29] introduced the concept of a virtual flexible hinge that was quasi-statically equivalent to the actual parallel-hinge configuration, and the behavior of parallelogram mechanisms with straight-axis hinges and stage devices with circular-axis hinges was analyzed. Grames et al. [30] presented a compact 2 degree-of-freedom crossed cylinders wrist mechanism suitable for automatically controlled surgical operations. As for the design of multi-functional compliant mechanisms, an electrically conductive LET joint was explored in Ref. [31], which includes two conductive layers between three insulated layers. A conductive hinge could be used in foldable back-packable solar arrays and can alleviate common issues associated with wire failure due to fatigue across joints. Dearden et al. [32] designed the inverted L-Arm gripper mechanism, it can overcome the disadvantages of using compliant mechanisms as a gripper, and the gripper can reach at least ± 90° of rotation. Ref. [33] made a robot using double-layer LEMs, which is driven by magnetic force. This paper further discussed how to adjust the motion by changing different parameters of the mechanism.

In this work, based on the design concept of improving the tensile stiffness of LET joints in Ref. [18], this paper proposed a DL-LET joint by fixing a layer of flexible H18 aluminum foil a LET joint. A theoretical model is derived for the DL-LET joint and verified by FEA and experimental results. The rest of the paper is organized as follows: Section 2 shows the design and analysis process of the DL-LET. Section 3 provides a performance comparison between DL-LET and LET joints. In Section 4, the performance of the DL-Triple-LET is compared with that of Triple-LET. Section 5 summarizes this paper.

## 2. Design and Analysis of DL-LET

### 2.1. Design of DL-LET

The schematic diagram of a DL-LET joint is shown in Figure 1. H18 aluminum foil has the advantages of large Young's modulus and high strength while maintaining good toughness. Beryllium bronze has obvious superiorities in terms of strength, mechanical property, and fatigue resistance. Therefore, the two materials are selected as layers in the DL-LET joint, respectively. Using strong welding adhesive, the LET joint of beryllium bronze is bonded to the H18 aluminum foil, which is consistent with the overall dimensions of the joint.

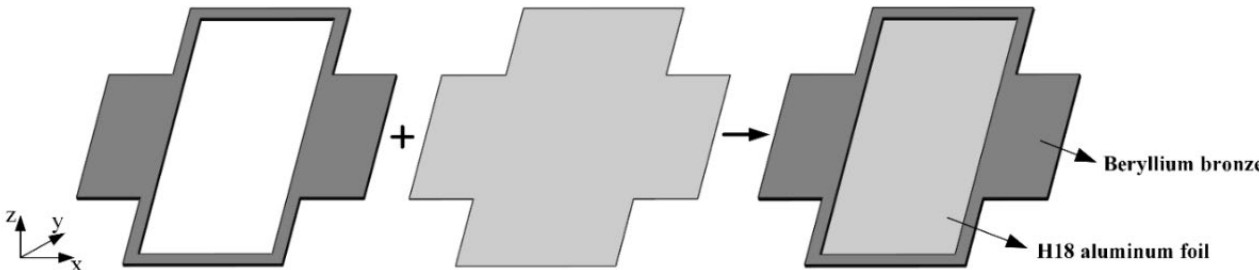

**Figure 1.** Schematic of a DL-LET joint.

The dimension of the DL-LET joint is shown in Figure 2. In the beryllium bronze layer, the length of the torsion segment is $l_{t1}$, and the width is $w_{t1}$; the length of bending segment 1 is $l_{b1}$, and the width is $w_{b1}$. In the H18 aluminum foil, the length of bending segment 2 is $l_{b2}$, and the width $w_{b2}$ is the same as $l_{b1}$. The thicknesses of the beryllium bronze layer and the H18 aluminum foil layer are $t_1$ and $t_2$, respectively.

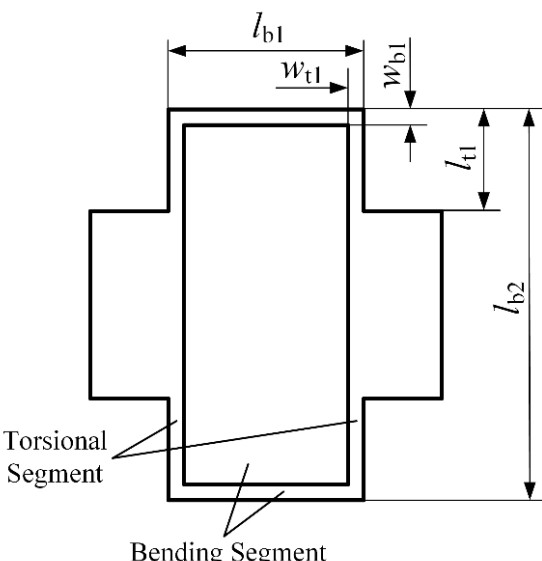

**Figure 2.** Dimension of a DL-LET joint.

### 2.2. Stiffness Modeling

In order to find the equivalent spring constant of DL-LET joints ($k_{\mathrm{eq,bend}}$), the DL-LET joint can be simplified as a spring system. Since the H18 aluminum foil is soft and its thickness is much smaller than beryllium bronze, when considering the bending stiffness of the DL-LET joint, the H18 aluminum foil is regarded as a bending segment in parallel with the LET joint. Figure 3 shows the linear spring model of the DL-LET joint.

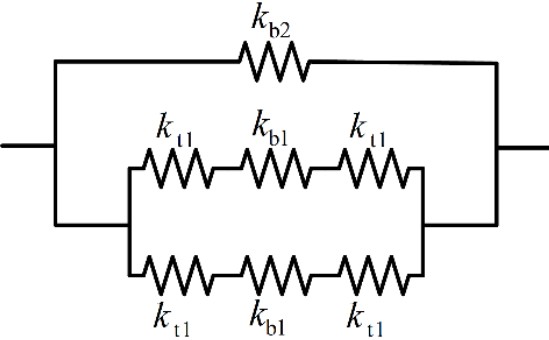

**Figure 3.** DL-LET joint equivalent to the corresponding spring series and parallel relations.

The equivalent stiffness of the DL-LET joint can be derived from the series-parallel connection relationship of the spring, thus $k_{\text{eq,bend}}$ is,

$$k_{\text{eq,bend}} = \frac{k_{\text{t1}}k_{\text{b2}} + 2k_{\text{t1}}k_{\text{b1}} + 2k_{\text{b1}}k_{\text{b2}}}{2k_{\text{b1}} + k_{\text{t1}}} \tag{1}$$

In Equation (1), $k_{\text{b1}}$ and $k_{\text{b2}}$ represent the equivalent spring stiffness of the bending segments, respectively, and $k_{\text{t1}}$ represents the equivalent spring stiffness of each torsion segment.

As for bending segments, the deformation can be regarded as the maximum deflection produced by the free end of the cantilever beam under torque, so each bending segment's equivalent spring constant can be found by,

$$k_{\text{b1}} = \frac{E_1 I_{\text{b1}}}{l_{\text{b1}}} \tag{2}$$

$$k_{\text{b2}} = \frac{E_2 I_{\text{b2}}}{l_{\text{b2}}} \tag{3}$$

where $E_1$ and $E_2$ refer to Young's modulus of beryllium bronze and H18 aluminum foil, respectively, and $I_{\text{b1}}$ and $I_{\text{b2}}$ represent the moment of inertia.

As for torsion segments, each spring constant can be determined by,

$$k_{\text{t1}} = \frac{KG_1}{l_{\text{t1}}} \tag{4}$$

in which $K$ is a coefficient related to the cross-sectional geometry of the torsion segments [34]:

$$K = \frac{1.17w_t^2 + 2.191tw_t + 1.17t^2}{w_t^2 + 2.609tw_t + t^2} \times \frac{2t^3 w_t^3}{7t^2 + 7w_t^2} \tag{5}$$

$G_1$ is the shear modulus,

$$G_1 = \frac{E_1}{2(1 + \nu_1)} \tag{6}$$

and $\nu_1$ is the Poisson's ratio. Equation (5) is thickness-to-width ratio independent and suitable for variable cross-section beams and optimization design of torsional elements in compliant mechanisms [24].

The H18 aluminum foil layer can be seen as a thin flat material, and it plays a major role during the tensile process. With the increase of tensile, the deformation of DL-LET in the *x*-direction is minimal to be ignored.

### 2.3. Analysis of DL-LET Joint

An FEA model of the DL-LET joint is created to evaluate the joint behavior and verify the correctness of the theory. A 3D model is established and analyzed with the Abaqus

finite element software. NLGEOM is turned "On" in the step analysis to include the nonlinear effects of large displacement. In selecting two surface interaction types, the binding constraint is used to realize that the relative motion and deformation do not occur between the two surfaces. The joint is fixed at one end, and the free end is applied a bending moment. The FEA results show that the running results converge.

Considering the feasibility of processing, the design dimensions are as follows: $l_{b1}$ = 25 mm, $w_{b1}$ = 2 mm, $l_{t1}$ = 13 mm, $w_{t1}$ = 2 mm, $l_{b2}$ = 50 mm, $w_{b2}$ = 25 mm, the thickness of beryllium bronze layer $t_1$ is 0.5 mm, and the thickness of H18 aluminum foil $t_2$ is 0.01 mm, with Young's modulus $E_2$ = 30 GPa, Poisson's ratio $\nu_2$ = 0.32. The equivalent stiffness of the DL-LET joint can be obtained using Equations (1)–(6):

$$k_{\text{eq,bend}} = 118.77 \text{ N·mm·rad}^{-1} \tag{7}$$

The stress distribution in the joint is shown in Figure 4 when a bending moment of 50 N·mm is applied. The H18 aluminum foil is so thin that the stress is negligible during deflection. The largest stress occurs in the beryllium bronze layer. At this point, the bending angle of the joint is 0.414 rad, and the maximum equivalent stress of the beryllium bronze layer is 306.0 MPa, which is smaller than the yield limit of beryllium bronze $[s_y]$ = 1170 MPa. The maximum equivalent stress of H18 aluminum foil is 113.5 MPa, which is smaller than the yield limit of H18 aluminum foil, $[s_y]$ = 124 MPa. Therefore, the plastic deformation will not occur and meet the design requirements.

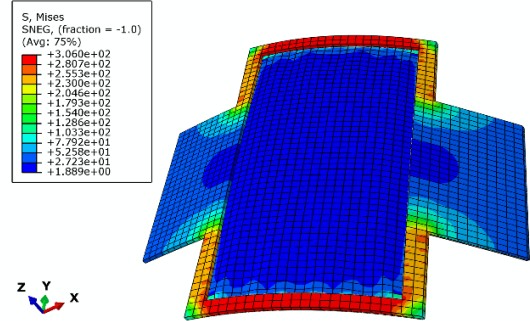

**Figure 4.** Stress plot.

When the torque exerting on the joint is $T$, we can acquire the following equations of the relationship between $T$ and $\theta$:

$$T = k_{\text{eq,bend}}\theta \tag{8}$$

where $\theta$ is the rotation angle of the joint.

The theoretical rotation angles of the DL-LET joint under different bending moments can be obtained by Equation (7), as plotted in Figure 5.

Under different bending moments, the comparison between the theoretical and simulation values of the rotation angles is shown in Figure 5a, and the errors are shown in Figure 5b. It can be seen from Figure 5a that the theoretical values are consistent with the simulation values, showing an excellent linear relationship. It can be seen from Figure 5b that with the increase of bending moment, the relative error decreases slightly, and all errors are within 3%, which proves the correctness of the theoretical calculation method.

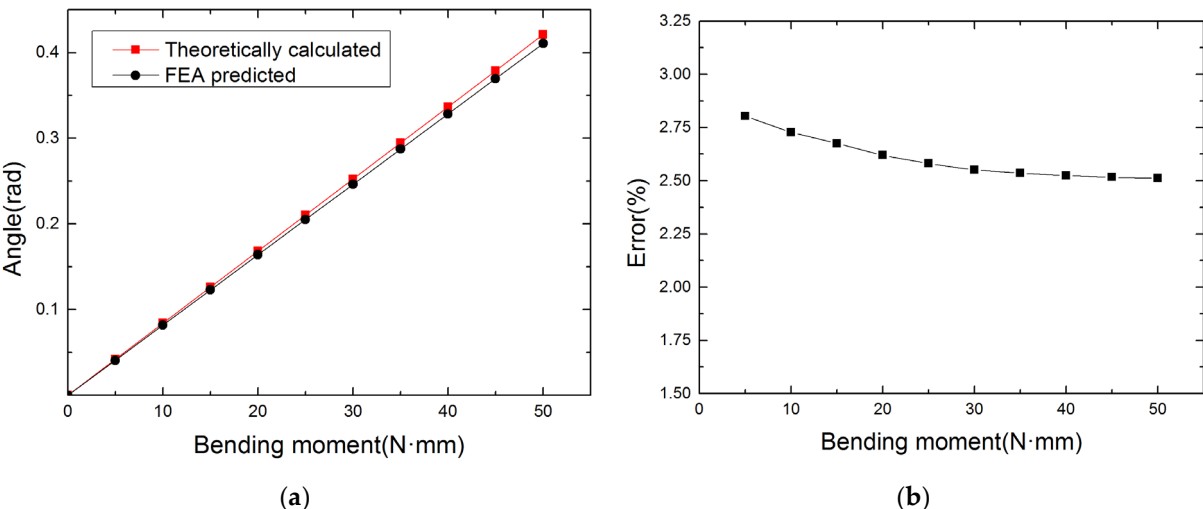

(**a**) (**b**)

**Figure 5.** (**a**) Angle versus bending moment curves and (**b**) Error versus bending moment curve (rotation angle).

## 3. Performance Comparison between DL-LET and LET Joints

### 3.1. Validation by Finite Element Results

As a baseline, the analysis of the LET joint was made to compare bending and tensile stiffness with the DL-LET joint. The material for the LET joint is assumed to be beryllium bronze, the overall dimensions are the same as the DL-LET joint, and others remain the same.

Applying a bending moment of 50 N·mm and a tensile load of 30 N on two joints, the FEA results are shown in Figures 6 and 7, respectively. DL-LET and LET joints have a similar bending stiffness, while the tensile deflection of the benchmark LET joint is 7.1 times larger than that of the DL-LET joint. The bending angles and tensile deflections as loads functions are plotted in Figure 8. It can be found that the H18 aluminum foil significantly improves the tensile stiffness of the DL-LET joint without affecting its bending stiffness.

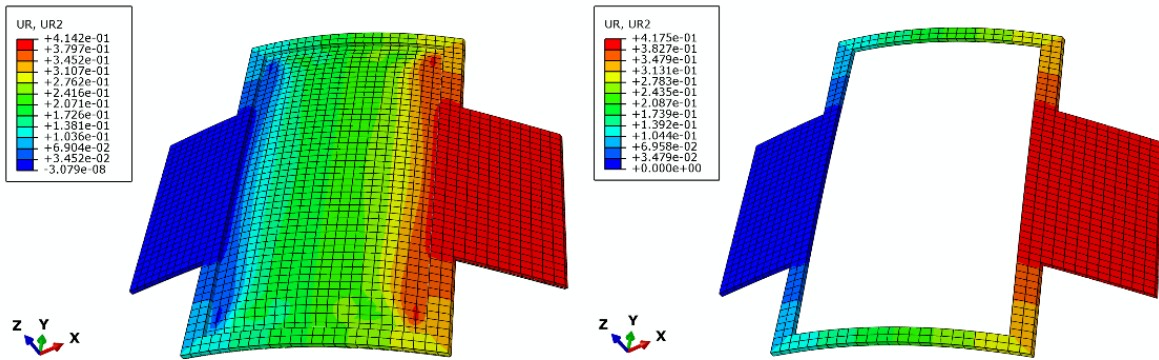

**Figure 6.** Bending deformations of DL-LET and LET joints obtained by FEA.

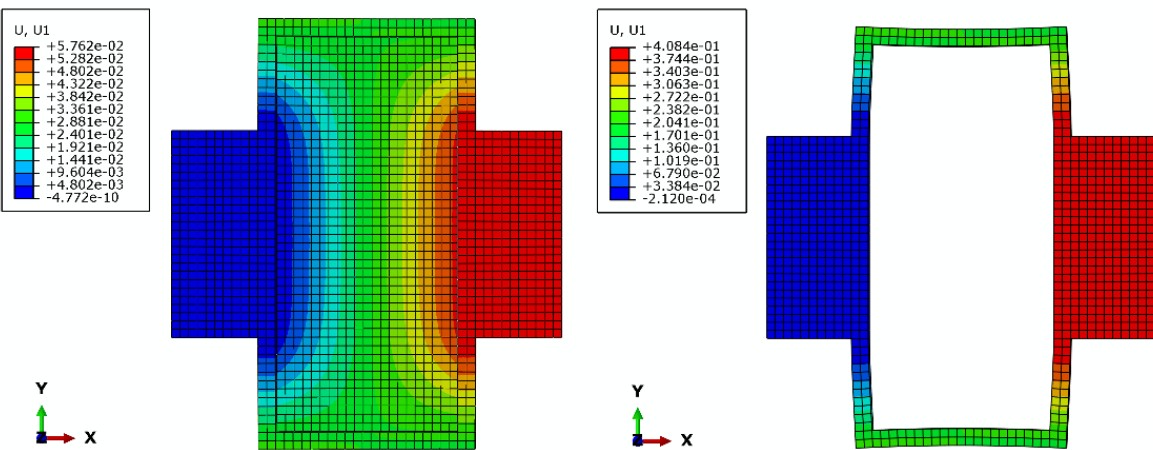

**Figure 7.** Tensile deformations of DL-LET and LET joints obtained by FEA.

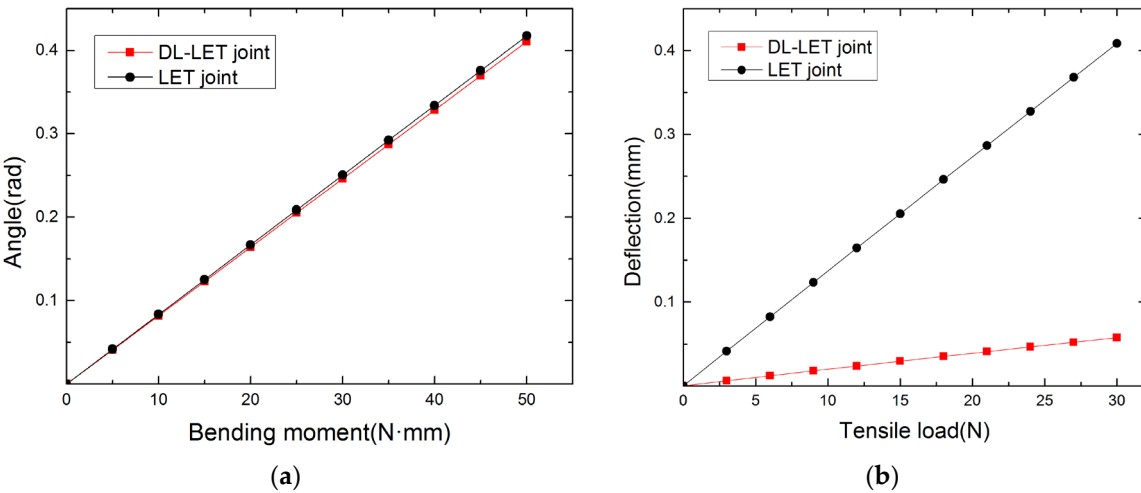

**Figure 8.** Comparison of FEA results of DL-LET and LET joints, (**a**) Bending deformation and (**b**) Tensile deformation.

### 3.2. Testing

The LET joint and DL-LET joint prototypes were fabricated to verify the performance of the proposed designs, as shown in Figure 9, and on the left is the DL-LET joint. The measured dimensions of the LET and the DL-LET joints are 24.97 mm × 49.89 mm and 24.96 mm × 49.90 mm, respectively. On a Universal Testing Machine, clamping the joints were clamped on both ends and stretched at the speed of 0.01 mm/min, and the force sensor recorded the tensile force. The experimental setup is shown in Figure 10.

When the tensile load is 30 N, the measured deflections of the LET and the DL-LET joints are 0.432 mm and 0.072 mm, respectively. It can be concluded that the deformation of LET is about 6.0 times larger than that of the DL-LET. The deflections of the two joints under different tensile loads are compared in Figure 11. The discrepancy between the experimental and FEA results could be attributed to bonding imperfections between the beryllium bronze and H18 aluminum foil layers. Furthermore, the combined effects of various factors will affect the measurement results, such as manufacturing error, equipment error, etc. Although the experimental results are not as perfect as the FEA results, the trend of experimental results is in line with the expected design requirements.

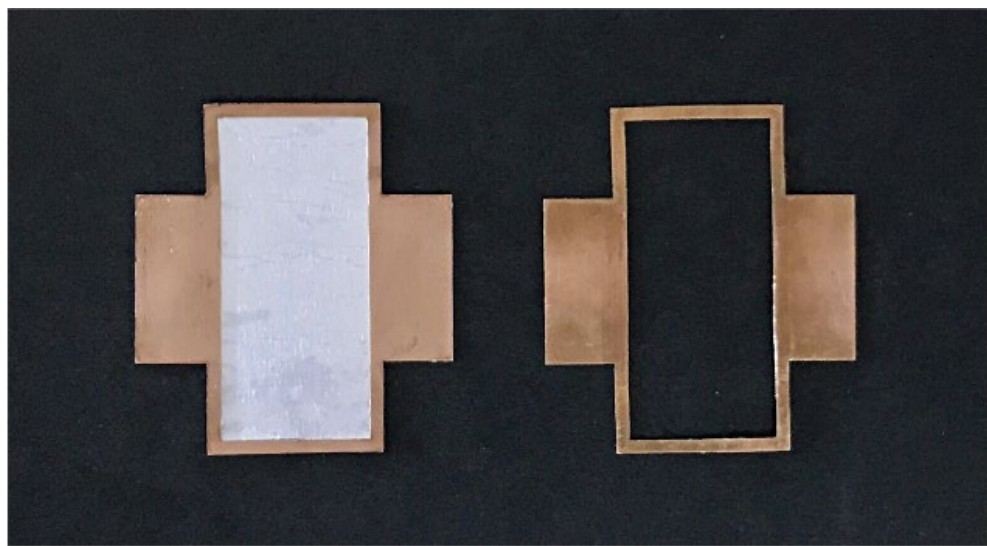

**Figure 9.** Prototypes of a DL-LET joint and a LET joint.

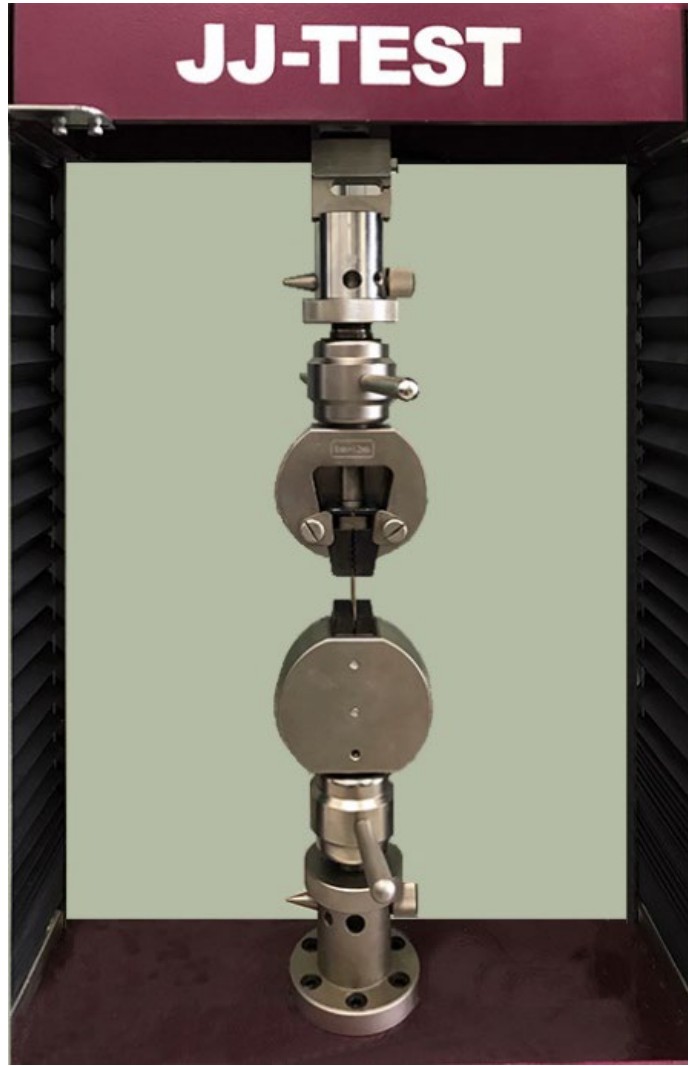

**Figure 10.** Testing Machine.

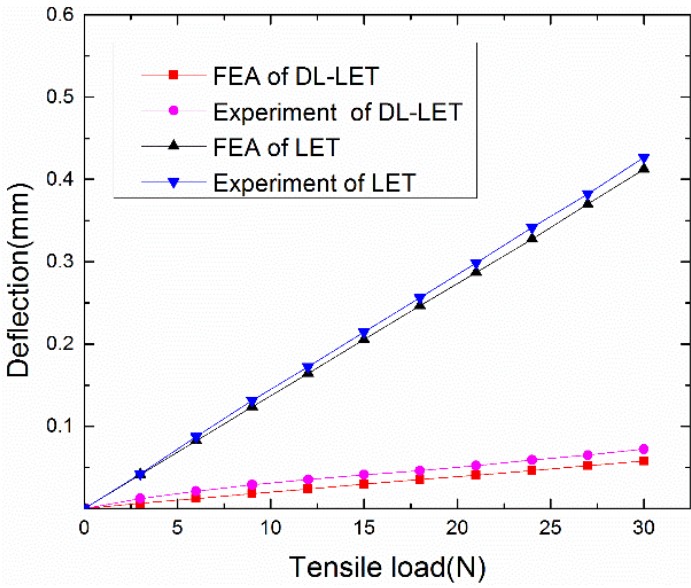

**Figure 11.** Comparison between DL-LET and LET joints.

## 4. Performance Comparison between DL-Triple-LET and Triple-LET

In order to further demonstrate the effectiveness of this idea and explain the effect of the double-laminated material structure on bending and tensile properties in compliant joints with different bending stiffness. Double-laminated Triple-LET (DL-Triple-LET) is designed using the method described in Section 2. It is shown in Figure 12.

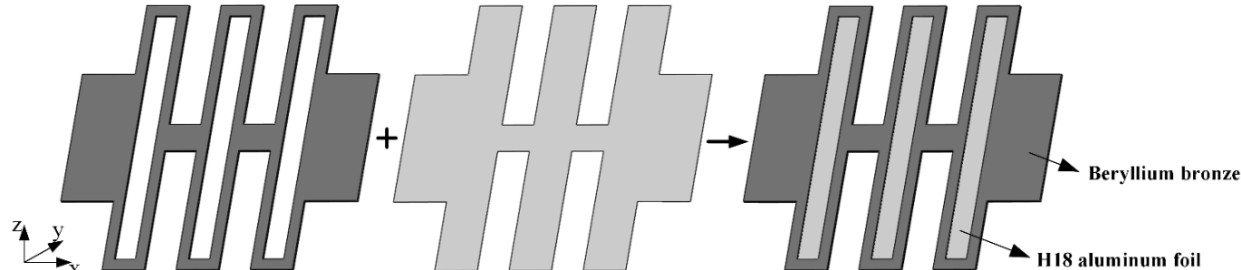

**Figure 12.** Schematic of a DL-Triple-LET joint.

A Triple-LET joint consists of three LET joints connected in series. Thus, it can achieve larger deformation [15]. At the same time, it is poor in anti-tensile property, and it will give rise to parasitic motion, which makes the motion accuracy of the joint is limited.

DL-Triple-LET joint's structure is based on the configuration proposed in Ref. [15], and the dimensions are redesigned. The Triple-LET joint's dimensions are shown in Figure 13 and Table 1, and the material is defined as beryllium bronze. The beryllium bronze layer of the DL-Triple-LET joint has the same dimensions as the Triple-LET joint. The H18 aluminum foil layer with a thickness of 0.01mm has the same overall dimensions with the same dimensions Triple-LET joint.

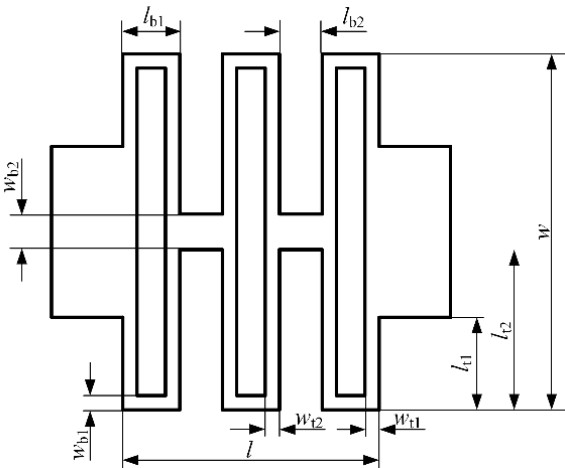

**Figure 13.** Dimension of a Triple-LET joint.

**Table 1.** Geometric parameters of the Triple-LET joint.

| Dimension | Value (mm) | Dimension | Value (mm) |
|---|---|---|---|
| $l_{b1}$ | 4 | $l_{b2}$ | 3 |
| $w_{b1}$ | 1 | $w_{b2}$ | 8 |
| $w_{t1}$ | 2 | $w_{t2}$ | 1 |
| $l_{t1}$ | 21 | $l$ | 18 |
| $w$ | 50 | $t$ | 0.5 |

Using Abaqus, the deformations of the DL-Triple-LET and Triple-LET joints subject to a bending moment of 50 N·mm and are shown in Figure 14, and the deformation of the DL-Triple-LET and Triple-LET joints with a tensile load of 30 N is shown in Figure 15. It can be seen from Figures 14 and 15 that the value of deflection of Triple-LET is 28.8 times larger than that of the DL-Triple-LET joint, while the bending stiffness is almost the same. Figure 16 shows the trends of the simulation values of the two joints under different bending moments and tensile loads.

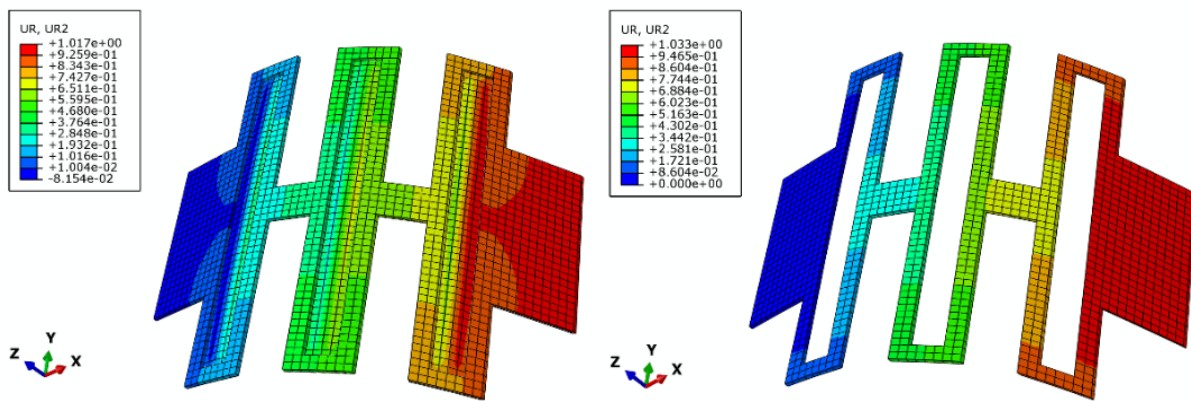

**Figure 14.** Bending deformations of DL-Triple-LET and Triple-LET joints obtained by FEA.

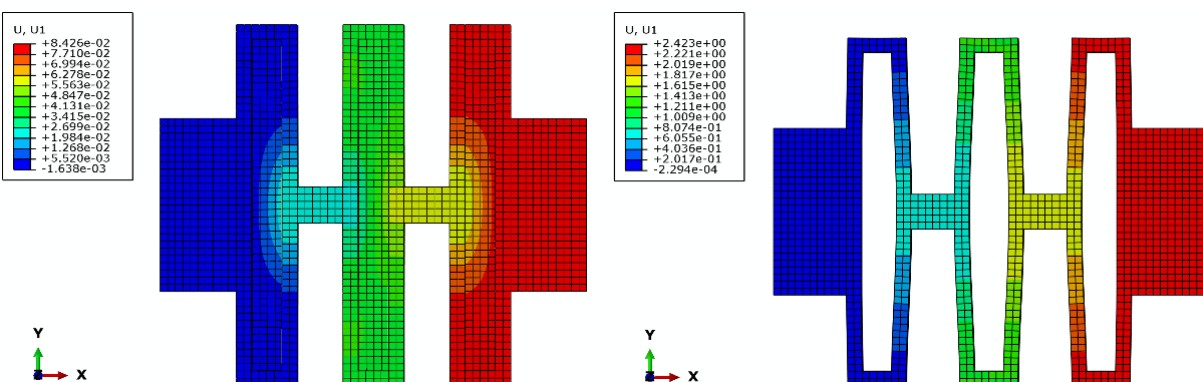

**Figure 15.** Tensile deformations of DL-Triple-LET and Triple-LET joints obtained by FEA.

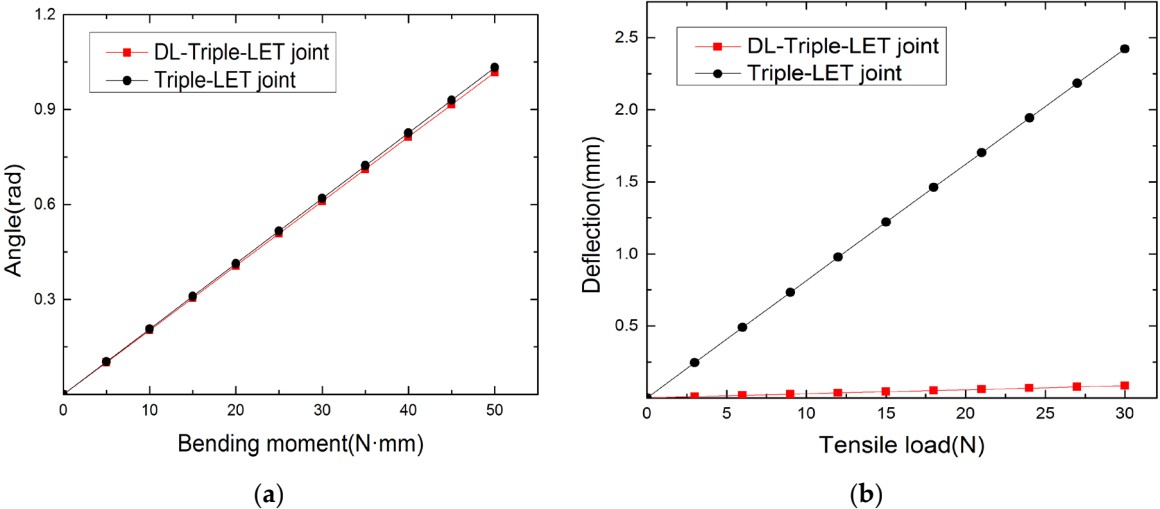

(**a**)                                                                                    (**b**)

**Figure 16.** Comparison of FEA results of DL-Triple-LET and Triple-LET joints, (**a**) Bending deformation and (**b**) Tensile deformation.

Compared to the DL-LET joints in Section 2, the improvement of tensile stiffness in DL-Triple-LET was more significant. It can be concluded that the double-laminated material structure plays a more critical role in joints with a smaller bending stiffness for improving the tensile stiffness.

## 5. Conclusions

The DL-LET joint, which utilizes a layer of flexible H18 aluminum foil to increase the tensile stiffness of the LET joint, was proposed. The kinetostatic model for the DL-LET joint was given, and the equivalent bending stiffness was derived. The model was verified using finite element analysis (FEA). The results showed that DL-LET and LET joints have similar bending stiffness, while the tensile stiffness of DL-LET joints is several times larger than those of LET joints, which was further validated by tensile tests. Finally, the DL-Triple-LET joint was presented, and its performances were compared to the Triple-LET joint. The DL-LET joint is vulnerable to delamination of the two layers.

Furthermore, this idea can also be used to combine different materials, such as more flexible materials such as polypropylene, and can also be used for a combination of multiple materials. The idea of combing two materials provides a new way to compliant joint design. Meanwhile, it provides an effective way to improve the tensile stiffness of joints. For future work we will develop variable stiffness techniques of this joint, and explore its application in a robot arm considering both kinetostatic and transient behaviors.

**Author Contributions:** This paper has six authors. The authors made most of the contributions regarding conceptualization, development of theory, validation, verification of the analytical methods, discussion of the results, as well as the final manuscript. Individual contributions are as follows: introduction, methodology, visualization, and original draft preparation: B.M., N.L. and Y.Y.; review, editing, and formal analysis: B.L., L.Q. and G.C.; software, validation: N.L. and Y.Y.; project administration, supervision, and funding acquisition, L.Q. All authors have read and agreed to the published version of the manuscript.

**Funding:** The authors would like to thank the support provided by the National Natural Science Foundation of China (Nos. 51475037 and U1913213).

**Informed Consent Statement:** Informed consent was obtained from all subjects involved in the study.

**Data Availability Statement:** Data will be available through the corresponding authors: guimin.chen@xjtu.edu.cn (Guimin Chen).

**Conflicts of Interest:** The authors declare no conflict of interest.

## Nomenclature

| | |
|---|---|
| **LET joint** | Lamina Emergent Torsional joint |
| **DL-LET** | Double-laminated LET |
| **FEA** | Finite element analysis |
| **LEMs** | Lamina emergent mechanisms |
| **I-LEJ** | Inverted Lamina Emergent joint |
| **T-LEJ** | Tension Lamina Emergent joint |
| **IT-LEJ** | Inverted tension Lamina Emergent joint |
| **CA** | Compliant arrays |
| **OD-LEJ** | Outside-Deployed Lamina Emergent Joint |
| **M-LET** | Membrane-enhanced LET joints |
| **NLGEOM** | Geometric nonlinearity |
| **DL-Triple-LET** | Double-laminated Triple-LET |
| **Triple-LET** | Three LET joints connected in series |
| **BCM** | the Beam constraint model |
| **CSBCM** | the Spatial Chained Beam-Constraint-Model |
| $k_{eq,bend}$ | the equivalent stiffness of the DL-LET joint |
| $k_{b1}$ and $k_{b2}$ | the equivalent spring stiffness of each bending segment |
| $k_{t1}$ | the equivalent spring stiffness of each torsion segment |
| $l_{t1}$ | the length of the torsion segment in the beryllium bronze layer |
| $w_{t1}$ | the width of the torsion segment in the beryllium bronze layer |
| $l_{b1}$ | the length of the bending segment in the beryllium bronze layer |
| $w_{b1}$ | the width of the bending segment in the beryllium bronze layer |
| $t_1$ | the thicknesses of the beryllium bronze layer |
| $E_1$ | the Young's modulus of beryllium bronze |
| $I_{b1}$ and $I_{b2}$ | the moment of inertia of each segment |
| $K$ | the coefficient related to the cross-sectional geometry of the torsion segments |
| $G_1$ | the shear modulus of beryllium bronze |
| $\nu_1$ | the Poisson's ratio of beryllium bronze |
| $w_{b1}$ | the width of the bending segment in the H18 aluminum foil |
| $l_{b2}$ | the length of the bending segment in the H18 aluminum foil |
| $t_2$ | the thicknesses of the H18 aluminum foil layer |
| $E_2$ | the Young's modulus of H18 aluminum foil |
| $\nu_2$ | the Poisson's ratio of H18 aluminum foil |
| $\theta$ | the rotation angle of the joint |
| $T$ | the torque exerting on the joint |

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
