# Peer review of "Design and Performance Analysis of Lamina Emergent Torsional Joints Based on Double-Laminated Material Structure"

_applsci, doi:10.3390/app12052642_

Round 1

Reviewer 1 Report

Nice piece of research but one aspect needs to be addressed. Following the reported physical experiments (Fig. 10 + description) steady state inverstigation is acceptable. Transient analysis followed by reliability experiments/analysis/modeling of the proposed structure would be interesting and critical in aspect of joint application contrains. In my opoinion is should be added or explained in context of scheduled follow-up activities.

Author Response

Thanks for your encouragement and suggestion. We agree that transient analysis is important in designing a robot joint. However, the transient analysis requires relevant parameters of the robot arm be determined besides the parameters of the joint. The following sentence has been added at the end of Conclusions:

"For future work we will develop variable stiffness techniques of this joint, and explore its application in a robot arm considering both kinetostatic and transient behaviors"

Reviewer 2 Report

Chen, G.; Ma, F.; Hao, G.; Zhu, W. Modeling large deflections of initially curved beams in compliant mechanisms using 277 chained Beam-Constraint-Model, ASME J. Mech. Robot., 2019, 11: 011002, https://doi.org/10.1115/1.4041585.

The article being evaluated is an extension of the above cited reference, and the authors would be well advised as to summarize this article, in part, and to delineate what has changed as of 2019, i.e. put this in a paragraph and to explain the subsequent evolution since 2019.

Secondly what is missing in the article is in terms of application a clear delineation as to what benefits robotics accrue by the developments summarized in this article . What new capacities and abilities so created by this evolutionary jump

Aside from that the article is solidly researched and is useful as a source of technology development

Author Response

Thanks a lot for your encouragement and suggestion.

The mentioned reference is focused on an analytical method for modeling large deflections in compliant mechanisms, while this work presents a method of enhancing the tensile strength of lamina emergent joints by using double-layer materials. As compared to Ref. [19] which focuses on concepts and demonstrations, this work mainly focuses on analytical modeling of one design concept and provides finite element and experimental validations. The following sentences have been added at the beginning of the last paragraph of Introduction: "In this work, based on the design concept of improving the tensile stiffness of LET joints in Ref. [19], this paper proposed a DL-LET joint by fixing a layer of flexible H18 aluminum foil a LET joint. A theoretical model is derived for the DL-LET joint and verified by FEA and experimental results."

As to the second comment, Ref. [34] has been new added in the reference list and the following sentence has been added at the end of the second paragraph of Introduction: "Ref. [34] made a robot using double-layer LEMs, which is driven by magnetic force. This paper further discussed how to adjust the motion by changing different parameters of the mechanism". The following sentence has been added at the end of Conclusion: "For future work we will develop variable stiffness techniques of this joint, and explore its application in a robot arm considering both kinetostatic and transient behaviors."

Reviewer 3 Report

The flow of the paper or the organization of the paper has to be presented at the end of the introductory section.
Abstract (what you are going to do) & conclusions (what you have done) are not sharp, rewrite it.
A part of the abstract looks like it has been copied from standard papers.
Keywords are not sufficient after the abstract.
I think, the similar work has been done by many people, which has to be referenced, i.e., the base paper from where the work has started has to be written (The base paper is not mentioned or cited).
Work done by various authors has not been cited properly in the manuscript.
Recent references are to be added as well as it has to be cited, majority of them are old references.
References are not written in the standard journal format.
The recent work done by the author of the paper in the relevant field also has to be mentioned in the references.
Similarity index is there with many of the papers already published, hence, use of plagiarism software is recommended to test for the originality before submitting the final manuscript.
Many of the paragraphs seems to be copied from standard papers (to be changed in your own words in simple English).
Work relevant to the research topic is not incorporated, i.e., the work done by other researchers / authors in this field are not incorporated.
The disadvantages, drawbacks of the works in the other author’s papers in the relevant field are not put.
The future work in this field is not mentioned.
Literature survey is inadequate; incorporate the recent works of the authors.
There is lot of spelling mistakes & grammatical errors, which has to be corrected using spell check & by other means.
Paragraphs are lengthy & have to be cut into smaller sizes.
Flow chart or the algorithm of the proposed work should be incorporated.
Figures have to be in .jpeg (insert mode).
Paper is not in the relevant conference / journal format; redo it according to the format specified by the committee.
The existing work is not compared with some standard technique, if this point is implemented, then the proposed work would be authenticated that it is a better concept than the others, if not done, so, please do it & add it so that it will be a full-fledged paper which can be cited by many.
The paper has to be revised giving the proper flow of reading, It seems like abruptly written.
English writing is very poor, some abbreviations are used without telling what it is, It will hamper the flow of reading.
Figures & tables should be cited in the text & the insertion of the same should be very near to where it is cited.
Nomenclature has to be put at the end along with the acknowledgement.
Paper contains the content, but it has to be revised in a standard way.

Author Response

Thanks a lot for your encouragement and suggestion.

We have rewritten the conclusions and abstract, and added additional keywords.

The following sentences have been added in the first paragraph of Introduction to mention the base paper and the related work (more references have been added): "Finding suitable compliant joints [7] can be a key step in designing LEMs. Jacobsen et al. [8] first proposed the design concept of utilizing torsional deflection to obtain out-of-plane motions from planar structures, leading to a new group of compliant joints for LEMs called Lamina Emergent Torsional (LET) joints". At the beginning of the last paragraph of Introduction: "In this work, based on the design concept of improving the tensile stiffness of LET joints in Ref. [19], this paper proposed a DL-LET joint by fixing a layer of flexible H18 aluminum foil a LET joint. A theoretical model is derived for the DL-LET joint and verified by FEA and experimental results."

The references have been reformatted according to standard journal format.

We have added more relevant references in this version.

Reviewer 4 Report

The current paper proposes a novel lamina emergent torsional (LET) joint named double-laminated lamina emergent torsional joint (DL-LET) joint which utilizes a layer of flexible H18 aluminum foil to increase the tensile stiffness of LET joint. The theory was validated using simulations.

Comments to authors:

- Please add more details regarding paper’s novelty, it is not very clear what are the novelties of this paper.

- Please add more details of how the theory from the first sections is applied in the results section and better detail the theoretical part, in this version the theoretical part is almost inexistent.

- Please detail how the parameters were obtained.

- The state of the art it is very poor regarding representative papers, maybe the author could add the following publications:

o Hybrid Data-Driven Fuzzy Active Disturbance Rejection Control for Tower Crane Systems, European Journal of Control, vol. 58, pp. 373-387-11, 2021.

o Event-Triggered Adaptive Fuzzy Control for Stochastic Nonlinear Systems with Unmeasured States and Unknown Backlash-Like Hysteresis, IEEE Transactions on Fuzzy Systems, doi 10.1109/TFUZZ.2020.2973950, pp. 1–19, 2020.

- Please add more details regarding the obtained results.

- Add the both the advantages and the disadvantages of the proposed method. In the proposed manuscript only the advantages are presented.

Author Response

We thanks the reviewer for the encouragement and valuable comments.

1. The contribution of this work with respect to the relevant work is further described in Introduction (as marked in the manuscript).

2. The theoretical part has been extended in the modeling section (page 3).

3. The modeling section (page 3) has been extended to make it clear how the parameters were obtained.

4. The recommended and other relevant references have been added.

5. Subsection 3.2 has been rewritten to add more details regarding the obtained results.

6. The following sentence has been added at the end of first paragraph of Conclusions to clarify disadvantage of DL-LET: “The DL-LET joint is vulnerable to delamination of the two layers.”

Round 2

Reviewer 1 Report

Thank you for your explanation.

Reviewer 4 Report

The authors answered to all my concern in the second version of the paper. In conclusion the paper deserves to be acceped as contribution in Applied Science journal.